# Addictive Behaviors, Depression, and Quality of Life among Korean Fishermen

**DOI:** 10.3390/healthcare11111648

**Published:** 2023-06-05

**Authors:** Mi Yeul Hyun, Suyoung Choi

**Affiliations:** College of Nursing, Health and Nursing Research Institute, Jeju National University, Jeju 63243, Republic of Korea; hpeople@jejunu.ac.kr

**Keywords:** fishermen, addictive behaviors, alcohol, gambling, depression, quality of life

## Abstract

This study investigated addictive behaviors (alcohol dependence and gambling tendencies), depression, and quality of life (QoL) among Korean fishermen in the Jeju Island region, Korea. The study utilized the Alcohol Use Disorder Identification Test—Korean version, the Korean version of the Canadian Problem Gambling Index, the Center for Epidemiological Studies Depression Scale, and the Korean version of the World Health Organization QOL-BREF to measure the study variables. The results showed that 18.1% of the fishermen had alcohol dependence and 9.9% abused alcohol, 13.6% were categorized as problem gamblers, 15.2% were moderate risk gamblers, and 14.4% were low-risk gamblers; 25.1% and 20.8% suffered from severe and mild depression, respectively. The mean QoL score was 3.13 ± 0.56, and the psychological health section scored the highest. The degree of alcohol dependence varied by age, education level, and job satisfaction; gambling tendency varied by age, job position, and job satisfaction; depression varied by religion and job satisfaction; QoL varied by religion and job satisfaction. Alcohol dependence, gambling tendency, and depression were significantly negatively correlated with QoL. Specifically, higher levels of alcohol dependence were associated with lower QoL scores in the subcategories of physical health and psychological health, while higher levels of gambling tendencies were associated with lower QoL scores in the subcategories of physical health, psychological health, social relationships, and general subcategories. Finally, higher levels of depression were associated with lower QoL scores across all five subcategories. Overall, participants exhibited remarkably elevated levels of alcohol dependence, gambling tendencies, and depression, and lower QoL compared with the general population. Further efforts are required to increase Korean fishermen’s job satisfaction to improve these problems. In addition, public health policies must address and promote fishermen’s QoL.

## 1. Introduction

Individuals’ quality of life (QoL) is important. QoL is defined as one’s personal evaluation of the position they occupy relating to their personal goals, expectations, norms, and interests within the cultural context and value system that one belongs. This is a highly broad concept that is widely influenced by physical health, mental state, independence, and social and environmental relationships [1]. In other words, QoL represents the degree of happiness and life satisfaction experienced and evaluated by every individual. The aspects that comprise QoL are highly diverse, and measurement methods vary. To measure QoL, it would be most desirable to include all the variables of every single area suggested in previous studies. However, this study mainly measured QoL and variables related to mental health status.

Fishery is an activity with significant economic and social values, and it contributes to people’s well-being by providing fishery products. In Korea, however, the fishery industry is decreasing, and the income in fishing households is not showing signs of improvement. Moreover, fishery resources are also decreasing due to environmental changes and overfishing. Consequently, there is diminished hope about life in fishing villages [2]. Moreover, fishermen are at an elevated risk of suffering physical and mental health problems due to a poor working environment [3]. Unfortunately, few studies have addressed their health problems or QoL.

Addiction problems are an area of growing interest for researchers. Alcohol use disorder is one example, which is associated with many physical, mental, and social problems. The treatment of alcohol use disorder is also difficult due to the perceived social stigma, which is associated with treatment avoidance, a lack of knowledge, and frequent relapse. These problems affect not only the QoL of people who abuse alcohol but also the entire society [4]; therefore, mental health experts must devise an efficient approach to resolve this problem.

Furthermore, in Korea, there has been an increased trend of gambling and gambling addiction problems since 2000. Gambling addiction reduces QoL and causes household problems, such as bankruptcy and divorce [5]. Fishing and rural villages are not free from these phenomena, and many fishermen often pass the time by drinking or gambling. In a previous study conducted with fishermen who had attended a training seminar, 12.3% were categorized as problem gamblers [6]. Considering that the prevalence of gambling addiction among the general population is 5.1% [7], this rate is concerning.

Depression also affects QoL. Severe depression can cause a loss of interest, reduced sensory perception, such as the impaired ability to taste, smell, hear, and touch, and a lack of motivation in daily functioning and future planning. Moreover, recovery and adaptation capacity may be lost and self-esteem decreases. Therefore, depression is one of the most important variables that affect QoL [8].

Many studies have addressed QoL relating to diverse social classes and variables; however, few studies have examined fishermen. Korean studies of fishermen have been limited to issues such as fatigue and musculoskeletal diseases; studies addressing mental health levels or QoL are nearly non-existent. Therefore, the aim of this study was to assess the alcohol use, gambling tendencies, depression levels, and QoL of Korean fishermen. Additionally, this study aimed to examine the differences in these factors based on sociodemographic characteristics and determine the QoL levels based on alcohol use, gambling tendencies, and depression levels among Korean fishermen.

## 2. Materials and Methods

### 2.1. Study Design and Population

This was a cross-sectional, descriptive study conducted in the Jeju Island region of Korea. Jeju is inhabited predominantly by fishermen involved in fish-related occupations. The target population of this study was fishermen engaged in the fishing industry in Korea, and the accessible population was fishermen working in the fishing industry within the Jeju Island region. According to the 2010 national agriculture and fishery census, Jeju has 2506 fishermen: 58.0% reside in Jeju and 42.0% reside in Seogwipo. Five-hundred participants were sampled through quota sampling based on the 2010 national agriculture and fishery census results. Additionally, a power analysis was conducted to estimate the required sample size, which was determined to be 280 with a 95% confidence interval, an effect size of 0.25, and a test power of 0.95 at a 0.05 level of error. The authors visited training sessions or reunions of fishermen at 8 major ports located in both Jeju and Seogwipo to conduct the survey. A total of 410 copies were returned; however, 403 were used for analysis (7 were omitted because of missing data).

### 2.2. Measures

#### 2.2.1. Alcohol Dependence

For the identification of alcohol dependence, the Alcohol Use Disorder Identification Test—Korean version (AUDIT-K) [9], which was modified into Korean based on the original AUDIT developed by the World Health Organization (WHO) [10] (https://www.who.int/publications/i/item/WHO-MSD-MSB-01.6a, accessed on 15 May 2023), was used for measurement. This tool consists of 10 questions measured with a 5-point Likert scale (0–4; total score range 0–40), based on frequency of occurrence (e.g., 0 = never, 1 = less than monthly, 2 = monthly, 3 = weekly, and 4 = daily or almost daily). When the total score ranged between 0 and 7, respondents were categorized as low-risk drinkers (including non-drinkers); when the total score ranged between 8 and 15, 16 and 19, and 20 or more, participants were categorized as risky drinkers, alcohol abusers, and alcohol dependent, respectively. The internal consistency of this scale in this study was Cronbach’s α = 0.92.

#### 2.2.2. Gambling Tendency

To assess gambling tendency, we utilized the Korean Canadian Problem Gambling Index (KCPGI) [11]. The KCPGI is validated for use in the Korean population based on the original CPGI [12], which consists of nine items. The specific questions of the CPGI are as follows: (1) Have you bet more than you could really afford to lose? (2) Have you needed to gamble with larger amounts of money to get the same feeling of excitement? (3) Have you gone back another day to try and win back the money you lost? (4) Have you borrowed money or sold anything to get money to gamble? (5) Have you felt that you might have a problem with gambling? (6) Have you felt that gambling has caused you to develop health problems, including stress and anxiety? (7) Have people criticized your betting or told you that you have a gambling problem, whether or not you thought it was true? (8) Have you felt that your gambling has caused financial problems for you or your household? (9) Have you felt guilty about the way you gamble or what happens when you gamble? This scale employs a 4-point Likert scale (e.g., 0 = never, 1 = sometimes, 2 = most of the time, and 3 = almost always). When measuring gambling problems, 0 was categorized as non-problem gambling, 1–2 points was low-risk gambling; 3–7 points was moderate-risk gambling; and 8 points or higher was problem gambling. The internal consistency of this scale in this study was Cronbach’s α = 0.94.

#### 2.2.3. Depression

The Center for Epidemiologic Studies Depression (CES-D) Scale was first developed by Radloff [13]. The Korean version of the CES-D was translated by Cho and Kim [14], and it showed acceptable reliability and validity. The CES-D consists of 20 questions about symptoms that occurred in the week prior to the survey. Each item is scored from 0 (rarely or not at all) to 3 (most or all of the time), and is generally based on frequency of the symptoms. The total score ranges from 0 to 60 and higher scores indicate higher depression levels. Following the recommendations of Cho and Kim [14], this study categorized 21 points or higher as severe depression and 16–21 points as mild depression. The internal consistency of this scale in this study was Cronbach’s α = 0.82.

#### 2.2.4. QoL

QoL was measured with the Korean version of the WHO QOL-BREF [1]. It comprises 26 questions measured with a 5-point Likert design from 1 to 5. Higher scores indicate better QoL. The scales comprise 5 different categories: physical health (7 questions), psychological health (6 questions), social relationships (3 questions), environmental (8 questions), and general QoL. The internal consistency of the scale in this study was Cronbach’s α = 0.90; the subscales’ reliability ranged from α = 0.73–0.80.

### 2.3. Data Collection Procedures

The recruitment area comprised two trading ports, Jeju port and Seoqwipo port, as well as five coastal ports on Jeju Island, including Hallim port, Aewol port, Seongsanpo port, Hwasun port, and Chuja port. The data were collected during sailors’ meetings or training sessions. The inclusion criteria comprised the following: an adult aged 18 years or older, having nationality of the Republic of Korea, and being a fisherman who had sailing experience during the last year. Among these candidates, those who could understand and provide answers directly to the survey questions, who understood the study objectives, and who provided consent to participate were recruited for the study. Two study assistants, who had been trained for the study’s objective, survey contents, and the fishery characteristics, explained the purpose of the survey to participants, including the protection of their personal information and the fact that the participants could stop answering the survey at any moment. Then, after providing informed, written consent, the participants answered the survey.

### 2.4. Statistical Analyses

The data were analyzed using the Statistical Package for Social Sciences (SPSS) version 18.0. Descriptive statistics of participants’ general characteristics, alcohol dependence, gambling tendency, depression, and QoL were represented as means and standard deviations. To test for common method bias, Harman’s single-factor test was used, which can arise from the use of a single source of data. Harman’s single factor accounted for 18.8% of the total variance, which was less than the 50% threshold, indicating that common method bias was not a significant issue. Pearson’s correlation analysis was conducted to examine the relationship between alcohol dependence, gambling tendency, depression, and QoL. Additionally, independent t-tests and an analysis of variance were used to identify alcohol dependence, gambling tendency, depression, and QoL according to participants’ general characteristics, and QoL according to alcohol dependence, gambling tendency, and depression. Post hoc analysis was conducted through Duncan’s test.

### 2.5. Ethical Considerations

This study was approved by the Institutional Review Board of the University with which the researchers are affiliated (IRB No. 2013-07) and conducted in accordance with the Declaration of Helsinki. For data collection, a consent form including study purpose and contents, anonymity of data, and confidentiality was distributed to the participants prior to the survey questionnaire. Written consent was obtained from the participants who consented to study participation. Data were collected anonymously. The survey was typically answered by the participants themselves; however, if the participant was unable to read the survey and write the answers themselves, the authors completed the survey via a face-to-face interview. The survey took approximately 20 min to complete. A small gift was given to the participants in return for survey participation.

## 3. Results

### 3.1. Participants’ General Characteristics

Participants’ demographics are shown in Table 1. Most men were aged > 40 years, had > 10 years of education, were not religious, and were not satisfied with their job.

### 3.2. Participants’ Alcohol Dependence, Gambling Tendencies, Depression, and QoL

Participants’ levels of alcohol dependence, gambling tendencies, depression, and QoL are shown in Table 2. Most men were low-risk to risky drinkers, non-problem gamblers, with low levels of depressive symptoms. However, more than a quarter of the men either abused or were dependent on alcohol, were moderate-risk or problem gamblers, and nearly half the men had mild or severe depression.

### 3.3. Participants’ Alcohol Dependence, Gambling Tendencies, Depression, and QoL per Their General Characteristics

Alcohol dependence, gambling tendency, depression, and QoL were analyzed based on participants’ general characteristics (Table 3). For alcohol dependence, those in the age group of 41~50 and those with at least 10 years of education had higher levels compared with other groups, while those who were not satisfied with their job also had higher levels. For gambling tendency, those in the age group of 41~50, those in the job position of crew members, and those who were not satisfied with their job had higher levels compared with other groups. Depression levels were higher among those who did not identify as religious and those who were not satisfied with their job. Lastly, higher levels of QoL were observed among those who identified as religious and those who were satisfied with their job.

### 3.4. Relationship between Alcohol Dependence, Gambling Tendency, Depression, and QoL

The relationship between alcohol dependence, gambling tendency, depression, and QoL was examined (Table 4). Alcohol dependence showed a significant negative correlation with QoL mean score as well as a significant negative correlation with physical health, psychological health, and general QoL areas among the QoL sub-subcategories. Gambling tendency was significantly negatively correlated with the QoL mean score. Except for the environment QoL subcategories, it was significantly negatively correlated with physical health, psychological health, social relationships, and general areas. Depression also showed a significant negative correlation with QoL mean score and all five QoL subcategories.

### 3.5. QoL by Alcohol Dependence, Gambling Tendencies, and Depression

QoL by alcohol dependence showed differences in the subcategories of physical health and psychological health (Table 5). QoL by gambling tendencies showed significant differences with the QoL mean score and the subcategories of physical health, psychological health, social relationships, and general QoL (Table 6). QoL by depression levels showed significant differences in QoL mean score and all five subcategories (Table 7).

## 4. Discussion

This study examined the alcohol dependence, gambling tendencies, depression, and subsequent QoL of Korean fishermen.

The combined rate of alcohol dependence and abuse was much higher compared with a previously reported alcohol abuse rate (12.7%) [15], which was reported among non-fishermen Korean adults employing the same tool. However, the rate was lower than 32.4%, which was the alcohol dependence and abuse rate reported by a survey on manufacturing and public institution workers in Korea [16]. That study also showed a consistent result that the alcohol abuse rate by job was higher in office workers, management workers, or technicians/laborers compared with those who worked in agriculture and fisheries. Moreover, considering that the proportion of problem drinkers gradually decreases with age (aged 50 years and older) [16], it is expected that the rate of alcohol dependence and abuse will be lower in this study since almost half the participants were aged in their 50s.

The alcohol use rate by age group in this study showed that the use by men aged in their 40s was higher than that aged younger than 40 years or older than 50 years. This was consistent with Jang and Shin [16], who found that alcohol dependence of people aged in their 40s was the highest when analyzed by age group. Alcohol use level by education revealed that those holding a high school diploma or higher showed a higher level of alcohol use than those with a middle school diploma or lower. This corresponds in part with the result from a previous study [15] that showed that educated individuals who graduated from a university or higher had more alcohol problems compared with high school diploma holders or those who completed compulsory education. Therefore, alcohol problems seem to occur more frequently in highly educated groups.

In a study on health issues relating to the job [17], 51% of fishermen reported having job stress, which was the most commonly reported health issue. The second most reported issue was fatigue, accounting for 45.3%. Compared against the population of adults aged 19 years or older living in South Korea, who reported stress levels of 29.8% in 2010 and 30.6% in 2014 [18], the stress levels of the fishermen were strikingly high. On the other hand, considering that subjective stress has a significant influence on drinking [19], it is thought that the fishermen may have used alcohol to cope with high levels of stress and fatigue.

The prevalence of gambling addiction in Korea was 6.1% as of 2010, which is considerably higher compared with the U.K. (2.5%), France (1.3%), and Australia (2.4%) [20]. However, the gambling rate of the fishermen in this study was much higher, especially compared with the reported results of problem gamblers (1.3%) and moderate-risk gamblers (5.9%) in a 2012 gambling industry survey. One of the reasons for the serious gambling problem in Korea originates from a lack of quality leisure culture. The prevalent culture of playing games with gambling-like characteristics as part of leisure activities has led to an increased gambling tendency among individuals. A survey by Hyun and Cho [17] that examined the leisure activities of fishermen revealed that 58.8% of them spent time sleeping, 42.4% watching films or television series, 39.5% drinking, and 22.2% playing games.

Gambling is also known to reduce stress [21]. Considering the fact that complaints related to stress accounted for 51% among the health-related problems of fishermen, and a previous study showed that gambling levels increase as job stress worsens [17], policies to relieve fishermen’s stress and to promote diverse leisure activities may help remedy maladaptive coping mechanisms such as drinking and gambling.

Compared with the 2015 statistics published by Statistics Korea, which reported a depression incidence rate of 8.1% among Korean citizens, the depression rate in this study was remarkably higher. According to a report [22] based on the data from the 2014 National Health and Nutrition Survey, 11.6% of males were suffering from mild depression, and the prevalence of moderate, moderate-severe, and severe depression was 4.1% (3.2%, 0.6%, and 0.3%, respectively). Compared with these figures, the depression level in our study was extremely high.

The major factors that affect the depression level of workers are a lower income level and higher stress [23]. The work environment of fishermen is harsh and the income level is not rewarding; therefore, the fishermen’s high rate [17] of health complaints due to stress is thought to have affected their depression. Nonetheless, more profound analysis and more in-depth studies are required that address the factors influencing the depression level of fishermen.

Depression levels were also higher when job satisfaction was lower, which was consistent with a previous study [23]. While income is known to affect depression, no significant difference was found between income and depression in this study. It is likely that a trivial difference was observed between the groups since most earned below the national average (KRW 40 million as of 2012). Importantly, job satisfaction demonstrated significant differences in alcohol dependence, gambling tendencies, depression, and QoL. When it was low, alcohol dependence, gambling tendencies, and depression were more serious, and QoL was reduced.

The mean QoL score of the participants in this study was noticeably lower than the average QoL score of Korean adults (86.13) [1]. In fact, fishermen are known to demonstrate highly varied scores depending on the location of work or job type. Fishermen on the Latin American shores are much poorer compared with city inhabitants and their QoL is low. On the other hand, those working on the shores of Jalisco, Mexico, generally demonstrated a high QoL in previous studies. As these results suggest, the QoL of fishermen exhibits a wide range of differences [24]. Consequently, it is more logical to compare the QoL of Korean fishermen with the standard QoL score of Korean citizens rather than comparing it with fishermen in other countries. The QoL score showed about a five-point gap compared with the 72.48 points reported in a previous study [25], which studied similar participants of similar age groups (i.e., adults aged 40 years or older living in Seoul and Gyeonggi areas). Specifically, 93.1% of the fishermen responded that their level of living was average or below average, and 77.2% considered themselves to belong to the lower class [26]. The fact that they perceived their standard of living and subjective class consciousness to be lower can explain why their QoL score was low.

Concerning the subcategories of QoL, psychological health area scored highest, followed by physical health, general QoL, social relationships, and environment. This corresponds with the results from Park and Hyun [27]; however, their participants were engaged in distant-water fishing operations. Despite the differences in occupation, their QoL score and subcategory scores showed little difference.

QoL according to participants’ general characteristics was significantly higher when the fishermen were religious and when their job satisfaction was higher. This differed from previous results [6,27] that reported that QoL varied according to age and education. The general characteristics that influence QoL vary considerably from study to study; therefore, further investigation is required.

Alcohol dependence, gambling tendencies, and depression had a significant negative correlation with QoL, which corresponds with previous study results: QoL is reduced when alcohol problems are more serious [4], problem gambling and pathological gambling were significantly associated with low QoL [27], and depression was negatively correlated with QoL [28].

QoL according to alcohol dependence demonstrated significant differences in psychological health and physical health among the subcategories of QoL. This is related to a finding that alcohol-dependent patients demonstrated lower scores in psychological functioning than in physical functioning [29]. This further supports the notion that individuals with alcohol use disorder may benefit from psychological support.

QoL according to gambling tendencies showed significant differences in QoL mean score, and in physical health, psychological health, social relationships, and general QoL among the subcategories. It demonstrated significant differences in more areas than the QoL according to alcohol dependence. Consistently, a past study revealed that pathological gamblers showed lower QoL than individuals with alcohol use disorder, even though the former were younger, more educated, and employed, and the gravity of addiction was milder than the latter [30].

Lastly, QoL according to depression showed significant differences in QoL mean score and all five subcategories. This corresponds with a previous result that depression demonstrated the strongest correlation with QoL more so than any other variable [31]. Therefore, solving the problem of depression is a highly important challenge for healthcare practitioners.

## 5. Conclusions

This survey study investigated alcohol dependency, gambling tendencies, depression levels, and the consequent QoL of fishermen in the Jeju Island region of Korea. Fishermen who were aboard fishing vessels and performed tasks related to the vessels during the last year were surveyed. Too many of them were abusing alcohol and gambling and had prominent levels of depression and low QoL. Specifically, the QoL was much lower than the standard QoL score among Koreans, highlighting a grave reality for Korean fishermen. Based on these findings, preventative measures and treatment policies for alcohol and gambling problems among fishermen require establishment. Furthermore, the influential factors of depression should be more clearly identified as should the exploration of measures to improve Korean fishermen’s QoL.

There are several limitations of this study that should be considered when interpreting the results. Firstly, the participants were all recruited from a single region in Korea, so the generalizability of the findings to other populations may be limited. Secondly, the assessment of alcohol dependence, gambling tendencies, and depression was based solely on self-report measures and not on actual diagnoses from a clinician, which would have overestimated or underestimated the prevalence of these conditions among participants. Thirdly, the study relied on a single item to measure job satisfaction, which may not provide a comprehensive understanding of the relationship between job satisfaction and other variables among participants. Additionally, this study focused on identifying the levels of each variable. Future studies should attempt to identify the variables that influence each variable in the future. Therefore, future studies will be able to provide more concrete evidence for intervention measures to improve the QoL of fishermen. 

## Figures and Tables

**Table 1 healthcare-11-01648-t001:** Participants’ general characteristics. *N* = 403.

Characteristics	Categories	*N* (%) or M ± SD
Age (years)		49 ± 9
	≤40	66 (16)
	41–50	161 (40)
	≥51	176 (44)
Sex	Male	403 (100)
Education	Up to 9 years	140 (35)
	10 years or more	263 (65)
Spouse	Yes	324 (80)
	No	79 (20)
Religious	Yes	139 (34)
	No	264 (66)
Job position	Captain	178 (44)
	Engineer	65 (16)
	Crew	160 (40)
Job satisfaction	Not satisfied	242 (60)
	Satisfied	161 (40)
Annual income (unit: KRW 10,000)	<2000	106 (26)
	2000–3000	169 (42)
	>3000	128 (32)

Note: M ± SD (Mean ± Standard deviation).

**Table 2 healthcare-11-01648-t002:** Levels of alcohol dependence, gambling tendency, depression, and QoL. *N* = 403.

Variables	Categories	*N* (%)	M ± SD		Minimum	Maximum
Alcohol dependence	Low-risk drinking	171 (42)	2.82 ± 2.53	11.08 ± 9.41	0	40
	Risky drinking	119 (30)	11.08 ± 2.16			
	Alcohol abuse	40 (10)	17.23 ± 1.17			
	Alcohol dependence	73 (18)	27.04 ± 5.45			
Gambling tendencies	Non-problem	229 (57)	0	2.57 ± 4.57	0	27
	Low risk	58 (14)	1.38 ± 0.49			
	Moderate risk	61 (15)	4.26 ± 1.20			
	Problem gambling	55 (14)	12.67 ± 4.12			
Depression	Low	218 (54)	10.66 ± 3.10	15.89 ± 6.98	0	37
	Mild	84 (21)	17.81 ± 1.41			
	Severe	101 (25)	25.56 ± 4.03			
Quality of life	Physical			12.83 ± 2.53	5.14	20
	Psychological			13.16 ± 2.60	6.67	20
	Social			12.30 ± 2.62	4.00	20
	Environment			11.93 ± 2.85	4.50	20
	General			12.40 ± 3.19	4.00	20
	Mean score			3.13 ± 0.56	1.35	4.96

Note: M ± SD (Mean ± Standard deviation).

**Table 3 healthcare-11-01648-t003:** Variable analyses by participants’ general characteristics.

Characteristics	Categories	Alcohol Dependence		Gambling Tendency		Depression		Quality of Life	
		M ± SD	T or F (*p*)	M ± SD	T or F (*p*)	M ± SD	T or F (*p*)	M ± SD	T or F (*p*)
Age (years)	≤40 ^a^	10.14 ± 9.13	6.32 (0.002) (a, c < b)	1.71 ± 3.25	5.52 (0.004) (a, c < b)	15.61 ± 7.87	0.98 (0.378)	3.24 ± 0.61	1.41 (0.247)
	41~50 ^b^	13.08 ± 10.36		3.48 ± 5.63		15.40 ± 6.96		3.11 ± 0.57	
	≥51 ^c^	9.60 ± 8.26		2.07 ± 3.71		16.43 ± 6.64		3.12 ± 0.53	
Education	Up to 9 years	9.44 ± 8.06	−2.74 (0.006)	2.34 ± 4.20	−0.77 (0.444)	16.27 ± 6.70	0.809 (0.411)	3.09 ± 0.47	−1.13 (0.260)
	≥10 years	11.95 ± 9.96		2.70 ± 4.76		15.68 ± 7.13		3.15 ± 0.60	
Spouse	Yes	10.88 ± 9.26	−0.87 (0.387)	2.59 ± 4.55	0.15 (0.885)	15.69 ± 6.92	−1.13 (0.258)	3.14 ± 0.55	0.14 (0.889)
	No	11.90 ± 10.03		2.51 ± 4.69		16.68 ± 7.22		3.13 ± 0.60	
Religious	Yes	10.55 ± 8.58	−0.086 (0.393)	2.14 ± 4.01	−1.48 (0.141)	14.92 ± 6.16	−2.13 (0.034)	3.24 ± 0.59	2.72 (0.007)
	No	11.36 ± 9.83		2.80 ± 4.83		16.39 ± 7.34		3.08 ± 0.54	
Job position	Captain ^a^	10.06 ± 8.53	1.95 (0.143)	2.40 ± 4.23	3.40 (0.034) (b < c)	15.03 ± 6.81	2.43 (0.089)	3.21 ± 0.60	2.76 (0.065)
	Engineer ^b^	11.46 ± 9.83		1.51 ± 3.28		16.48 ± 6.04		3.10 ± 0.60	
	Crew ^c^	12.05 ± 10.09		3.19 ± 5.26		16.60 ± 7.44		3.07 ± 0.19	
Job satisfaction	Satisfied	9.55 ± 8.25	2.78 (0.006)	1.83 ± 4.10	2.76 (0.006)	13.77 ± 6.29	5.24 (<0.001)	3.35 ± 0.60	−6.46 (<0.001)
	Not satisfied	12.09 ± 10.00		3.07 ± 4.80		17.29 ± 7.07		2.99 ± 0.48	
Annual income (unit: KRW 10,000)	<2000 ^a^	12.77 ± 9.43	2.73 (0.067)	3.23 ± 5.05	1.87 (0.156)	16.76 ± 7.75	3.01 (0.051)	3.06 ± 0.57	2.79 (0.062)
	2–3000 ^b^	10.07 ± 9.26		2.14 ± 3.91		16.25 ± 6.66		3.11 ± 0.51	
	>3000 ^c^	11.01 ± 9.47		2.61 ± 4.92		14.68 ± 6.61		3.22 ± 0.56	

Note: M ± SD (Mean ± Standard deviation); ^a,b,c^ Duncan’s test (post hoc analysis).

**Table 4 healthcare-11-01648-t004:** Correlations between alcohol dependence, gambling tendency, and depression with quality of life.

Variables	1	2	3	4	5	6	7	8	9
	r (*p*)								
1. QoL mean score	1.00								
2. Physical QoL	0.84 (<0.001)	1.00							
3. Psychological QoL	0.84 (<0.001)	0.67 (<0.001)	1.00						
4. Social QoL	0.75 (<0.001)	0.56 (<0.001)	0.59 (<0.001)	1.00					
5. Environment QoL	0.86 (<0.001)	0.56 (<0.001)	0.55 (<0.001)	0.59 (<0.001)	1.00				
6. General QoL	0.74 (<0.001)	0.53 (<0.001)	0.61 (<0.001)	0.47 (<0.001)	0.60 (<0.001)	1.00			
7. Depression	−0.43 (<0.001)	−0.48 (<0.001)	−0.49 (<0.001)	−0.29 (<0.001)	−0.20 (<0.001)	−0.36 (<0.001)	1.00		
8. Gambling tendency	−0.16 (0.002)	−0.23 (<0.001)	−0.14 (0.004)	−0.13 (0.009)	−0.04 (0.422)	−0.13 (0.010)	0.30 (<0.001)	1.00	
9. Alcohol dependence	−0.10 (0.042)	−0.18 (<0.001)	−0.21 (<0.001)	−0.03 (0.611)	0.07 (0.163)	−0.11 (0.033)	0.39 (<0.001)	0.40 (<0.001)	1.00

Note: QoL: quality of life.

**Table 5 healthcare-11-01648-t005:** Quality of life by alcohol dependence. *N* = 403.

Quality of Life	Alcohol Dependence				F (*p*)
	Low-Risk Drinking ^a^ (*n* = 171)	Risky Drinking ^b^ (*n* = 119)	Alcohol Abuse ^c^ (*n* = 40)	Alcohol Dependence ^d^ (*n* = 73)	
	Mean ± Standard Deviation				
Physical	13.16 ± 2.55	13.07 ± 2.41	12.39 ± 2.59	11.90 ± 2.45	5.16 (0.002) (a, b > d)
Psychological	13.60 ± 2.61	13.20 ± 2.61	13.18 ± 2.32	12.05 ± 2.43	6.34 (<0.001) (a, b, c > d)
Social	12.32 ± 2.47	12.37 ± 2.80	12.33 ± 2.47	12.11 ± 2.81	0.16 (0.923)
Environment	11.72 ± 2.95	11.96 ± 2.74	12.25 ± 2.55	12.21 ± 2.98	0.70 (0.554)
General	12.63 ± 3.08	12.24 ± 2.98	13.20 ± 3.50	11.70 ± 3.49	2.45 (0.063)
Mean score	3.17 ± 0.56	3.15 ± 0.55	3.15 ± 0.54	3.01 ± 0.57	1.52 (0.210)

^a,b,c,d^ Duncan’s test (post hoc analysis).

**Table 6 healthcare-11-01648-t006:** Quality of life by gambling tendency. *N* = 403.

Quality of Life	Gambling Dependence				F (*p*)
	Non- Problem ^a^ (*n* = 229)	Low Risk ^b^ (*n* = 58)	Moderate Risk ^c^ (*n* = 61)	Problem Gambling ^d^ (*n* = 55)	
	Mean ± Standard Deviation				
Physical	13.16 ± 2.46	13.31 ± 2.40	12.60 ± 2.71	11.19 ± 2.11	10.53 (<0.001) (a, b, c > d)
Psychological	13.40 ± 2.69	13.60 ± 2.55	12.80 ± 2.38	12.07 ± 2.18	4.96 (0.002) (a, b > d)
Social	12.53 ± 2.66	12.51 ± 2.37	11.98 ± 2.90	11.47 ± 2.23	2.90 (0.035) (a, b > d)
Environment	12.13 ± 2.91	11.80 ± 2.45	11.61 ± 3.13	11.58 ± 2.70	0.95 (0.418)
General	12.65 ± 3.18	12.62 ± 3.21	12.36 ± 3.15	11.20 ± 3.07	3.20 (0.023) (a, b, c > d)
Mean score	3.20 ± 0.57	3.19 ± 0.52	3.06 ± 0.59	2.89 ± 0.44	5.26 (0.001) (a, b, > d)

^a,b,c,d^ Duncan’s test (post hoc analysis).

**Table 7 healthcare-11-01648-t007:** Quality of life by depression. *N* = 403.

Quality of Life	Depression			F (*p*)
	Low ^a^ (*n* = 218)	Mild ^b^ (*n* = 84)	Severe ^c^ (*n* = 101)	
	Mean ± Standard Deviation			
Physical	13.69 ± 2.50	12.36 ± 2.26	11.34 ± 1.99	37.16 (<0.001) (a > b > c)
Psychological	14.10 ± 2.53	12.57 ± 2.15	11.61 ± 2.19	41.24 (<0.001) (a > b > c)
Social	12.93 ± 2.60	11.59 ± 2.38	11.52 ± 2.53	14.74 (<0.001) (a > b, c)
Environment	12.36 ± 2.70	11.32 ± 2.87	11.53 ± 3.03	5.48 (0.004) (a > b, c)
General	13.16 ± 3.28	11.76 ± 3.03	11.31 ± 2.66	14.66 (<0.001) (a > b, c)
Mean score	3.31 ± 0.55	2.99 ± 0.50	2.87 ± 0.48	28.61 (<0.001) (a > b, c)

^a,b,c^ Duncan’s test (post hoc analysis).

## Data Availability

The data analyzed during the current study are available from the corresponding author on reasonable request.

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
