# Peer review of "Addictive Behaviors, Depression, and Quality of Life among Korean Fishermen"

_healthcare, 2023, doi:10.3390/healthcare11111648_

Round 1
Reviewer 1 Report (Previous Reviewer 3)
Thank you for revising the manuscript based on the suggestions.
Reviewer 2 Report (Previous Reviewer 1)
The authors reviewed the manuscript by implementing all the suggested changes, and I believe the overall quality of the reported study has improved. I don't have any other comments or suggestions.
This manuscript is a resubmission of an earlier submission. The following is a list of the peer review reports and author responses from that submission.
Round 1
Reviewer 1 Report
Thank you for the opportunity to review this manuscript. The authors report the results of a cross-sectional study in which they investigated the prevalence of addictive behaviors (alcohol and gambling), depression, and the perception of quality of life in a sample of Korean fishermen.
The manuscript is well-written and the methodology is simple (mainly descriptive analyses, without an attempt to explain how the variables influence each other, but this is acknowledged in the limitations) but correct, the results are clearly presented and the conclusions are coherent with the results.
I have just a couple of minor points which I encourage the authors to address to improve the overall quality of their manuscript.
- Please check the reference numbers in sections 2.2.1 and 2.2.2 (e.g., I believe that the references [9] and [10] on page 2 “For the identification of alcohol dependence, the Alcohol Use Disorder Identification-Korean version (AUDIT-K) [9] which was modified into Korean based on the original AUDIT developed by the World Health Organization (WHO) [8]” should be [10] and [9] instead; same in the 2.2.2 sub-section);
- It is not clear to me how job satisfaction was measured: was it a simple satisfied/not satisfied question? The authors should justify the usage of this simple question over the many job satisfaction measures available in the literature, and possibly add this as a limitation of the study;
- Another limitation of this study is that alcohol dependency, gambling tendencies and depression were assessed through the use of self-report measures only, not actual diagnoses from a clinician;
- The study adopted a cross-sectional methodology with self-report measures, therefore I suggest checking for common method bias using Harman's single-factor test.
Reviewer 2 Report
I was going to say that I am in support of publication. It is a very well-written article. The only three notes I had for the authors were: 1) Delete the first "changes" in line 40 so that it reads "due to environmental changes and overfishing." 2) Insert a comma after "port areas" on line 121 so that it reads "...small-scale port areas, including the major ones..." 3) Move 4. Discussion on line 198 to the next page so that there is not a line break between this and the beginning of the discussion
well written
Reviewer 3 Report

In general, the English language used in the manuscript is good. Minor edits are required to ensure clarity and avid confusion. Please use inclusive language; more information can be found here: https://www.apa.org/about/apa/equity-diversity-inclusion/language-guidelines
Reviewer 4 Report
The manuscript is original. Nevertheless, some amendments are required:
- The abstract is descriptive ponly, give more details on the Results.
- The Aim of the study needs to be better specified: what is (are) the hypothesis to be tested?
- Sample size calculations are not given
- The statistical paragrap is lacking of clarity and completeness. The correlation is not explained. Moreover, a multivariate analysis is completely missing. A linear regression analysis and òlogistic regression analysis are required, according to the dependent variable chosen.
- The results of the multivariate analysis need to be discussed in details in the Discussion section
Round 2
Reviewer 4 Report
The authors made the requested changes
Author Response
Thank you for your review and encouraging comments.